# Therapeutic Targeting of the Anaplastic Lymphoma Kinase (ALK) in Neuroblastoma—A Comprehensive Update

**DOI:** 10.3390/pharmaceutics13091427

**Published:** 2021-09-08

**Authors:** Annette K. Brenner, Maria W. Gunnes

**Affiliations:** Department of Paediatrics, Haukeland University Hospital, N-5021 Bergen, Norway; maria.winther.gunnes@helse-bergen.no

**Keywords:** neuroblastoma, anaplastic lymphoma kinase, point mutations, monotherapy, combinatory therapy, drug resistance

## Abstract

Neuroblastoma (NBL) is an embryonic malignancy of the sympathetic nervous system and mostly affects children under the age of five. NBL is highly heterogeneous and ranges from spontaneously regressing to highly aggressive disease. One of the risk factors for poor prognosis are aberrations in the receptor tyrosine kinase anaplastic lymphoma kinase (ALK), which is involved in the normal development and function of the nervous system. ALK mutations lead to constitutive activation of ALK and its downstream signalling pathways, thus driving tumorigenesis. A wide range of steric ALK inhibitors has been synthesized, and several of these inhibitors are already in clinical use. Major challenges are acquired drug resistance to steric inhibitors and pathway evasion strategies of cancer cells upon targeted therapy. This review will give a comprehensive overview on ALK inhibitors in clinical use in high-risk NBL and on the potential and limitations of novel inhibitors. Because combinatory treatment regimens are probably less likely to induce drug resistance, a special focus will be on the combination of ALK inhibitors with drugs that either target downstream signalling pathways or that affect the survival and proliferation of cancer cells in general.

## 1. Introduction

Precision medicine is an expanding field within cancer treatment. Malignancies are increasingly divided into subcategories depending on their genetic features. A variety of genetic alterations (such as ABL [1], IDH2 [2] and PD-L1 [3]) can already be specifically targeted, usually as a supplement to standard treatment regimes.

In neuroblastoma (NBL), amplifications and point mutations in the anaplastic lymphoma kinase (ALK) gene correlate with an adverse prognosis. In malignancies harbouring ALK mutations, the kinase is among the most promising targets for personalized therapy.

This review provides an update on ALK inhibitors administered alone or in combination with other targeted therapies in the treatment of NBL. The article summarizes the contents of recent publications on ALK inhibitors (especially [4,5,6,7]) but further includes results from recent clinical studies and novel research articles. In particular, the review gives a summary of the results of research studies focusing on ALK inhibition in combination with other (pathway-) specific inhibitors.

## 2. Neuroblastoma

### 2.1. Aetiology, Prevalence and Characteristics

Neuroblastoma (NBL) is an embryonic malignancy originating from multipotent migrating crest cells [8], specifically the sympathoadrenal progenitors and chromaffin cells [9,10,11]. During the normal development of the peripheral nervous system, these cells undergo epithelial-to-mesenchymal transition [12], whereafter they migrate as maturated cells to their respective tissues [11,13]. NBL probably arises from aberrant migration, maturation, and differentiation of these crest cells [13]. NBL lesions can appear along the entire sympathetic nervous system [10,11], but thoracic and abdominal tumours are the most common [14], and the lesions are usually situated near adrenal glands [15]. The tumours are highly heterogeneous, consisting of both undifferentiated mesenchymal cells (MES) and (more) differentiated adrenergic cells (ADRN) [11].

NBL is the most common extracranial solid tumour in children [16], and although the malignancy amounts to approximately 8% of all paediatric cancers [10], about 15% of paediatric cancer fatalities are caused by NBL [13,17], making it the most deadly cancer type for children under the age of five [9]. The median age at diagnosis is 19 months [9,13], and malignancy rarely develops in children over the age of five [8]. Only 1–2% of NBL cases are germline tumours [10,13]; thus, NBL mainly originates from abnormal embryonic development.

Compared to other malignancies, NBL is highly heterogeneous, as it ranges from spontaneously regressing tumours to aggressive metastatic disease [18]: As many as 50% of NBL cases show metastasized disease at diagnosis [8,19]. This heterogeneity is thought to be an intrinsic characteristic of the primary tumour and not a progression from low- to high-risk over time [13].

### 2.2. Risk Factors, Tumour Staging and Treatment

In NBL, both prognosis and treatment protocols are based on intrinsic risk factors and tumour staging. Favourable prognosis is associated with early disease onset (<18 months), more differentiated tumours, and whole chromosome gains. Adverse risk factors, on the other hand, are (bone) metastasis and segmental chromosome losses or gains, especially 1p loss, 11q loss, and 17q gain. The most important risk factor affecting one single gene are *MYCN* amplifications [20,21].

The International NBL Risk Group Staging System (INRGSS) divides NBL into four distinct groups: localized tumours (stage L1), locoregional tumours with local tumour infiltration (stage L2), metastatic disease (stage M), and metastatic disease in children younger than 18 months of age, where the metastases are restricted to specific organs (stage MS) [22,23]. Because the lesions in the latter group can spontaneously regress, stage MS might reflect a multifocal rather than a metastatic disease [11].

Treatment regimens are according to risk stratification, which may vary between different countries/continents. In general, risk groups are based on patient age, the INRGSS system, tumour histology/differentiation, tumour ploidity and segmental chromosome aberrations, and the presence of *MYCN* amplifications [24]. Low-risk NBL has an excellent prognosis and is treated merely by observation or tumour resection. Intermediate-risk NBL also has a very favourable prognosis and is usually treated with tumour resection and chemotherapy [25]. Still, about 50% of NBL cases are characterized as high-risk at diagnosis [26], which explains the high overall fatality of NBL, as this group still has an overall survival (OS) below 50% [25]. High-risk patients commonly receive a high-intensive 3-step treatment, with tumour resection either in step 1 or step 2: (1) induction chemotherapy; (2) myeloablative consolidation chemotherapy followed by autologous stem cell transplantation; and finally, (3) post-consolidation therapy, including radiation therapy, immunotherapy, and treatment with retinoic acids [27]. Relapsed NBL has a very poor 5-year OS of approximately 20% [19], which mostly amounts to relapse in about 50% of high-risk patients [17]. However, the outcome after the first relapse is still favourable for the initially low- and intermediate-risk groups, especially if the relapse is locoregional. Nevertheless, relapsed high-risk NBL and either metastatic or refractory disease still have an abysmal prognosis [19].

As already mentioned, *MYCN* mutations and, predominantly, gene amplifications account for the most important single gene risk factor, and they lead to the overexpression of the *MYCN*-coded transcription factor N-Myc [10]. *MYCN* amplifications are the most common somatic mutations in NBL [25] and are correlated with chromosome 17q gain [8,10], tumour invasiveness [16,28], progressive disease [26], and tumour metastasis [8,17]. Finally, *MYCN* amplifications are also associated with mutations in another NBL oncologic driver, namely the receptor tyrosine kinase anaplastic lymphoma kinase (ALK). Increased ALK mRNA and protein expression, as result of ALK amplifications and point mutations, correlate with metastasis and decreased OS [29] and thereby represent a risk factor even for patients who are otherwise categorized as low- or intermediate-risk NBL [30]. This review will focus on the role of ALK in NBL and its potential as a drug target.

## 3. ALK in Health and Disease

### 3.1. Structure and Function of ALK

The tyrosine kinase ALK is a member of the insulin receptor tyrosine kinase superfamily and was first discovered in fusion with nucleophosmin [31]. ALK plays a role in the normal development and function of the central and peripheral nervous systems [32].

ALK is a protein comprising 1620 amino acids with a signal peptide at the N-terminal followed by the extracellular ligand-binding domain (residues 19-1038), a short transmembrane section, and finally, the C-terminal tyrosine kinase domain (residues 1060–1620) [7]. Figure 1 illustrates the simplified structure of the kinase, whereas Table 1 explains the mutation sites that are included in the figure.

ALK is considered an “orphan kinase” even though several not-ALK-specific–ligands have been identified: midkin, pleiotrophin, and FAM150A [17,33]. Ligand binding leads to receptor homo-dimerization and the subsequent auto-phosphorylation of the kinase domain at the tyrosine residues Y1278, Y1282, and Y1283. The receptor protein tyrosine kinase phosphatases β and ζ de-phosphorylate ALK again after ligand detachment [4]. The ATP-binding site itself is situated in a hinge region between the C-terminal lobe and the N-terminal lobe of the kinase domain [34].

ALK activates several downstream signalling pathways, most importantly the Janus kinase-signal transducer and activator of transcription protein (JAK-STAT), the RAS-mitogen-activated protein kinase (RAS/MAPK), and the phosphoinositide 3-kinase-Akt-mechanistic (former: mammalian) target of rapamycin (PI3K-Akt-mTOR) pathways [4]. Aberrant signalling through these pathways is, amongst other factors, associated with tumorigenesis [35,36]. Figure 2 gives an overview on the ALK downstream signalling pathways.

### 3.2. The Role of ALK in Tumorigenesis

ALK mutations are common in many adult cancers and are the most prevalent in non-small-cell lung cancer (NSCLC) and anaplastic large cell lymphoma (ALCL) [5]. ALK mutations are also present in paediatric malignancies and are the most common in ALCL, inflammatory myofibroblastic tumour (IMT), and NBL [37]. Since ALK is involved in the development of the nervous system, it might not come as a surprise that ALK mutations drive NBL.

The most common ALK mutations, both in adult and paediatric malignancies, are chromosomal translocations, and more than 20 ALK fusion partners have been identified so far [4]. An important point mutation is the so-called gatekeeper mutation ALK L1196M, which is in the centre of the ATP-binding pocket and that hinders ATP-competitive drugs from entering the ATP-binding site [7]. Other point mutations involved in drug resistance are ALK G1202R and G1269A [7]. However, neither these point mutations nor translocations are typical of NBL.

ALK amplifications are present in about 4% of NBL cases [25], and these lead to constitutive ALK expression [4] and tumorigenesis [6]. ALK and *MYCN* are adjacent genes on the short arm of chromosome 2 (loci 2p23 and 2p24, respectively) [16,38], and amplifications of ALK almost exclusively co-occur with *MYCN* amplifications [39]. ALK point mutations also frequently co-occur with *MYCN* amplifications [30,40]. ALK and *MYCN* synergistically cooperate in malignant transformation [29], and either protein (ALK or N-Myc) regulates the transcription of the other. N-Myc is a transcription factor [26], which—among others—regulates ALK transcription [32]. Activated ALK, on the other hand, elevates the activity of the *MYCN* promoter, thus driving N-Myc expression [41]. ALK that is activated by point mutations additionally stabilizes N-Myc [41,42].

ALK point mutations are more prominent than amplifications in NBL: the consensus of the incidence of ALK point mutations at diagnosis for sporadic NBL appears to be in the range of 8–10%, but numbers ranging from 6 to 12% are reported in the literature. In the 1–2% of germline NBL, ALK point mutations are typical features of the disease [43]. At relapse, the incidence of ALK point mutations is increased to about 25%, which is partly because ALK mutations are acquired during/after treatment [25] and also because ALK mutations are more common in high-risk NBL cases, which, in turn, are more likely to relapse [44]. Of note, ALK amplifications and ALK point mutations are mutually exclusive [30].

More than 35 ALK point mutations [5] at close to 30 mutation sites [7] have been detected in NBL: with exception of a few sites at the juxtamembrane, the mutations are situated in the kinase domain [7]. There are three germline point mutations that exist: ALK G1182A, R1192P, and R1275Q [6]. In somatic NBL, there are three hotspot gain-of-function mutations accounting for 85% of all ALK point mutations: 30% at F1174 (usually substituted by L, but also V, I, C and S), 12% at F1245 (mostly substituted by C, but also V, L and I), and 43% at R1275 (Q and, less commonly, L) [30]. These hotspot mutations keep ALK constitutively activated in a ligand-independent manner [30]. Substitutions at F1174 are the most potent drivers, as they increase ATP-affinity to the kinase to the highest degree. R1275Q, on the other hand, induces the lowest ATP-affinity increase [32]. Less frequent mutations include ALK I1171N and Y1278S [7], where the latter is one of the three ALK auto-phosphorylation sites. The mutation that is the most often associated with *MYCN* amplifications is F1174 [30], leading to N-Myc overexpression [45] and thus accounting for an exceptionally aggressive NBL subtype. The position and type of prominent ALK point mutations in NBL are shown in Figure 1 and Table 1.

The three hotspot gain-of-function mutation sites cluster in the hydrophobic core around the ATP-binding site [5,46] and stabilize the inactive state of the kinase [30]: both F1174 and F1245 are, together with F1098 and F1271, part of a four-phenylalanine motif that stabilizes the hinge region by aromatic ring stacking [46]. However, while the substitution of F to L at position 1174 does not induce structural changes in the hydrophobic region, substitutions at R1275 disrupt the hydrogen bonds of R1275 to both D1163 and D1276. Both R1275Q and R1275L further disrupt the hydrogen bonds to C1097 and Y1278, which is thought to increase the phosphorylation rate of the latter [46].

ALK gain-of-function mutations augment signalling via downstream pathways [5,10,18,26] and inhibit both cell cycle arrest and apoptosis [47]. F1174 substitutions especially upregulate STAT and Akt-phosphorylation, while R1275Q activates Akt and MAPK [48] (Figure 2).

Because ALK is overexpressed through amplification and is constitutively activated by the hotspot mutations in approximately 14% of de novo NBL (increased to 25% at relapse), the kinase is one of the most prominent targets in NBL, especially as the therapeutic inhibition of N-Myc has proven to be challenging [17]. Since ALK is only highly expressed during the development of the neonatal brain [30], ALK-specific inhibitors are likely to exert little toxicity [4].

The remainder of the review focuses on the different types of ALK inhibitors, both in monotherapy and in combinatory therapy, and their potential and limitations in the treatment of NBL.

## 4. ALK Inhibitors in Monotherapy

Most ALK inhibitors have been designed in order to target ALK rearrangements and fusion proteins, which are most commonly found in non-NBL malignancies. Almost all clinical trials on ALK inhibitors so far have concentrated on NSCLC and ALCL. From these studies, it is evident that ALK inhibitors in monotherapy are more efficient against fusions compared to point mutations and amplifications [11,44].

This section focuses on four generations of kinase type I inhibitors (Table 2) and on ongoing work with type I ½ and type II inhibitors. Finally, two drugs that indirectly target ALK will be discussed.

### 4.1. The ALK Inhibitor Pioneer Crizotinib

Crizotinib (PF-2341066), an aminopyridine-based inhibitor, binds the ATP binding pocket of the kinase domain and targets both ROS proto-oncogene 1 (ROS1) and the hepatocyte growth factor receptor (HGFR or c-Met) in addition to ALK [49]. One limitation that quickly becomes evident is that ALK acquires point mutations upon treatment, which render crizotinib ineffective. One of the acquired mutations is F1174L—one of the NBL ALK hotspot mutations that increases ALK affinity towards ATP [49]. ALK R1275Q is the point mutation with the highest sensitivity towards crizotinib, as it has the lowest ATP binding affinity of the three hotspot mutations [5,44]; regarding ALK amplifications, results on crizotinib effectiveness are conflicting [4,50]. A second concern with crizotinib is metastatasization to the central nervous system (CNS) because the drug cannot cross the blood–brain barrier [4,51].

Two clinical studies (phase I and II) including NBL patients have been completed, and one case report has been published:
In a phase I clinical study of eleven refractory NBL patients, one patient obtained complete remission, and an additional three patients obtained stable disease. Of these patients, two had an R1275Q germline mutation, while the two somatic point mutations were R1275L and F1174L. The remaining seven patients, among them one harbouring the ALK R1275Q mutation, showed disease progression [32];In a follow-up phase II clinical study, twenty relapsed/refractory NBL cases (no germline mutations, no point mutation at F1245) were included, and only three patients showed complete or partial response. All of these patients, in addition to two with stable disease, harboured the ALK R1275Q mutation [44];A case report included two patients with ALK overexpression. It is not stated whether this overexpression resulted from ALK amplifications or increased mRNA expression. Both patients showed complete response, but one patient relapsed after 32 weeks of crizotinib monotherapy. The other patient remained in remission without further treatment during the two months of follow-up [52].

Additionally, one clinical phase III study including high-risk NBL is currently recruiting. Patients with ALK mutations will receive a combination of crizotinib and standard chemotherapy. Additionally, a phase I study for patients with various malignant neoplasms is active, but only patients >15 years of age are included, meaning that few NBL patients will meet the inclusion criteria. All of the completed or ongoing clinical trials including NBL patients are listed in Table 3.

In conclusion, the first-generation ALK inhibitor crizotinib can stabilize disease or can lead to remission for a minority of patients, especially for those harbouring a mutation at R1275.

Due to the obvious limitations of crizotinib, novel drugs—most of them crizotinib derivatives—were designed.

### 4.2. A Wide Selection of Second-Generation of ALK Inhibitors

#### 4.2.1. TAE-684

The first second-generation ALK inhibitor was TAE-684, which, similar to crizotinib, is effective against the R1275Q mutation [6] but additionally is more effective against F1174L [48]. Furthermore, TAE-684, in contrast to crizotinib, inhibited downstream signalling pathways in TAE-684-sensitive cells [47,55]. However, TAE-684 never entered clinical trials due to its potential toxicity [7].

#### 4.2.2. Ceritinib/LDK-378

The most studied and used second-generation drug so far is ceritinib. It is a low molecular weight TAE-684 derivative. Ceritinib binds to the hinge-region of the kinase domain, which improves selectivity and potency compared to crizotinib [7], even though ceritinib has off-target effects on ROS1 and insulin-like growth factor receptor 1 (IGFR1) [4]. In contrast to crizotinib, ceritinib can enter the CNS and is effective against acquired post-crizotinib treatment mutations [56]. In cell cultures, ceritinib down-regulates the phosphorylation of ALK and of the downstream targets Akt and MAPK [49]. However, ALK F1174L/C mutations are resistant towards ceritinib [57,58], and overexpression of N-Myc—which is correlated with the presence of the F1174L mutation—contributes to drug resistance [49]. One phase I clinical trial on the use of ceritinib in relapsed or refractory NBL is currently recruiting, as is a phase II study including de novo NBL cases. A phase I ceritinib dose-escalation study in ALK-activated tumours has been completed: twenty-two patients were enrolled, seven of whom were NBL patients. Early analyses showed response in ALCL and IMT tumours; however, only one NBL patient, who was harbouring an ALK F1174L point mutation, showed partial response with tumour shrinkage in the retroperitoneum but with simultaneous disease progression in the CNS [53]. An additional case study reported more encouraging results: a 16-months-old patient with the novel ALK I1171T point mutation was treated with ceritinib monotherapy, which induced tumour cell differentiation and reduced tumour size, making it possible to resect the tumour. After 34 months of continuous ceritinib treatment, the child remained in complete remission [59].

#### 4.2.3. ASP3026

ASP3026 is the inhibitor that structurally is most closely related to TAE-684, with only a substitution in the aminopyrimidine ring. Similar to ceritinib, ASP3026 binds to the hinge-region of the kinase domain and additionally targets ROS1 and the activated CDC24 kinase (ACK) [54]. It has been shown to reduce tumour progression in a xenograft model [60]. In NSCLC, response towards crizotinib-resistant ALK mutations is comparable to ceritinib [54]. A phase I study on 12 patients, one of whom was a NBL patient, showed stable disease for the crizotinib-resistant ALK F1174L mutation [54].

#### 4.2.4. Brigatinib/AP26133

Brigatinib is a TAE-684/ceritinib analogue, which has off-target effects on ROS1 and the epidermal growth factor receptor (EGFR) [7]; brigatinib has shown to also inhibit activating mutations in the latter protein [61]. Brigatinib has not entered clinical trials on NBL but has shown promising results in in vitro and in vivo experiments. The drug can cross the blood–brain barrier, reducing the risk of CNS metastases [33]. Brigatinib reduces the phosphorylation of the tentative ALK auto-phosphorylation site Y1604 in addition to the downstream targets ERK5 and Akt [33]. In patient-derived xenograft (PDX) studies, brigatinib reduced tumour growth [33]. Most importantly, brigatinib was highly efficient against several NBL ALK mutations: the germline mutations G1182A, R1192P, and the somatic mutations R1275Q, Y1278S, and F1174L [33].

#### 4.2.5. Ensartinib/X-396

Ensartinib is a crizotinib-analogue that inhibited ALK phosphorylation in two cell lines and delayed tumour growth in a PDX model [62]. It seems to be efficient against crizotinib-resistant F1174L and R1275Q mutations [61]. Two clinical phase II trials for the use of ensartinib in recurrent/refractory NBL are currently recruiting.

#### 4.2.6. AZD3463

AZD3463 is a dual ALK-IGFR1 inhibitor that down-regulates proliferation and blocks the activation of the PI3K-Akt-mTOR signalling pathway, including for cases with the ALK F1174L mutation [63]. In this single study, AZD3463 increased apoptosis and autophagy and rendered cells more sensitive to doxorubicin. Of note, AZD3463 also reduces proliferation and increases apoptosis in cells with wt ALK [63]. It has not entered clinical trials in NBL.

#### 4.2.7. CEP-37440

CEP-37440 is the last second-generation inhibitor with an aminopyrimidine core. Its structure is loosely based on ceritinib but with a more kinase-selective profile [64]. It also targets the focal adhesive kinase (FAK), whose expression is indicative of multiple solid tumours [64]. CEP-37440 can enter the CNS and thus prevents brain metastases [64]. Even ALK F1174L seems to be sensitive towards CEP-37440, which has shown promising results in pre-clinical trials [64]. However, CEP-37440 has not been tested in NBL yet.

#### 4.2.8. Alectinib/CH5424802

Alectinib, a heterotetracyclic compound, is a highly selective second-generation ALK inhibitor that binds to the ATP-binding site [65] and that can pass the blood–brain barrier [66]. Alectinib inhibits the phosphorylation of the downstream targets Akt and MAPK and, importantly, also the expression of N-Myc [4,65]. Alectinib inhibits the three NBL ALK hotspot mutations, the germline mutations G1182A and R1192P, and the infrequent NBL ALK mutations I1171N, F1174V, and Y1278S [58]. Furthermore, ALK amplifications are sensitive towards alectinib [67]. In cell culture, alectinib leads to cell cycle arrest and apoptosis, whereas the inhibitor has an anti-tumoral effect in PDX models [67]. Alectinib has not entered clinical trials in NBL, but a case report on refractory NBL showed encouraging results. A 15-year-old patient with stage 3 NBL (according to the International Neuroblastoma Staging System) and an ALK F1245C point mutation relapsed 11 months after treatment with the NBL high-risk regimen. Partial response was achieved and maintained for 12 months with ceritinib monotherapy prior to a second relapse. Alectinib monotherapy reduced symptoms and resulted in a good partial response before a new relapse after 16 weeks [68].

#### 4.2.9. Belizatinib/TSR-011

Belizatinib is a selective ALK inhibitor with minor off-target effects towards the IGFR1/Janus kinase 2 (JAK2)/pan-tropomyosin receptor kinase (TrkA-C) and Src [7,69]. Specifically, ALK R1275Q and the NBL-uncommon gatekeeper mutation L1196M appear to be sensitive towards belizatinib [69]. Clinical trials on NSCLC patients show promising results [61].

#### 4.2.10. Entrectinib/RDX-101

Finally, entrectinib is an ALK/ROS1/pan-Trk inhibitor [70] that binds outside of the ATP-binding site [7]. Entrectinib inhibits proliferation, induces apoptosis, and reduces the phosphorylation of the downstream targets STAT and MAPK [17]. Entrectinib readily crosses the blood–brain barrier [4]. ALK amplifications and crizotinib-resistant mutations, not including the typical NBL point mutations, are sensitive to entrectinib [71]. Autophagy is an adverse side effect, especially in F1174L substituted ALK. However, this can be overcome by combining entrectinib with the autophagy inhibitor chloroquine, which even exerted synergistic effects in combination with entrectinib [71]. An observation from clinical studies, excluding those for NBL, is that entrectinib should be used as part of the first-line therapy because mutations acquired during/after crizotinib treatment are less responsive to entrectinib [72]. Despite entrectinib being a pan-Trk inhibitor, elevated levels of phosphorylated TrkB have been observed in entrectinib-resistant NBL mural models [73]. A phase I/II clinical trial including NBL patients is active.

### 4.3. Third-Generation of ALK Inhibitors

Thus far, two third-generation ALK targeting inhibitors have been FDA approved: lorlatinib and repotrectinib. Compared to the second-generation inhibitors, these novel compounds are made up of rigid 3D macrocycles with high affinity to the adenine binding site [7,15], where the structure of lorlatinib is a cyclic rearrangement of crizotinib.

#### 4.3.1. Lorlatinib/PF-06463992

Lorlatinib is a dual ALK/ROS1 inhibitor [50] that effectively targets most ALK mutations, including the three NBL hotspot mutations, in cell lines and PDX models [74,75]. Lorlatinib even inhibits the aggressive ALK F1174L mutation in combination with N-Myc overexpression [76]. Furthermore, lorlatinib can prevent CNS metastases as it readily passes the blood–brain barrier [74].

In contrast to the first and second generations of ALK inhibitors, treatment resistance in the case of lorlatinib is not driven by ALK (acquired) point mutations but is instead driven by the pathway evasion strategies of the cancer cells, namely through the PI3K-Akt-mTOR and the RAS/MAPK pathways. Additionally, resistance via EGFR and ErbB4 kinase evasion has been observed in cell lines [74]. However, the latter study was restricted to two NBL cell lines; thus, the results may not be representative for NBL in general.

One phase I clinical study on lorlatinib for treating de novo NBL with ALK amplifications or point mutations is currently recruiting, as is a phase I clinical trial for relapsed/refractory NBL with ALK amplifications or point mutations. For the latter study, only patients with prior treatment with first- or second-generation ALK inhibitors are eligible. Results from the dose escalation phase of this study have been published: Drug toxicity appears to be tolerable, and an objective anti-tumour response is evident, as eight of eighteen NBL patients experienced partial response or stable disease [77].

#### 4.3.2. Repotrectinib/TPX-0005

Repotrectinib is an “improved edition” of lorlatinib [7]. It is a kinase inhibitor targeting also ROS1 and pan-Trk [15]. Among the successfully targeted mutations are the three NBL hotspot mutations, the germline mutations G1182A and R1192P and the infrequent NBL ALK mutations I1171N and Y1278S. Of note, the usually drug-sensitive ALK R1275Q point mutation shows a comparably low response towards repotrectinib [15] and might therefore instead be targeted by another ALK inhibitor.

In cell line studies, repotrectinib decreased cancer cell proliferation while increasing apoptosis [15]. It further aborted the phosphorylation of the ALK downstream targets MAPK, Akt, and STAT [7,15]. In PDX models, repotrectinib showed a reduction in tumour size [15]. Repotrectinib has not entered clinical trials in NBL.

### 4.4. Improved Targeting Effects: Types I ½ and II Kinase Inhibitors

All of the inhibitors that have been presented thus far are type I kinase inhibitors. These are steric, competitive inhibitors that occupy the adenine binding pocket [78]. They therefore only target the active, i.e., ATP-bound, state of a kinase, thus obstructing auto-phosphorylation [5,17,26,78]. This offers the possibility for the acquisition of (point) mutations that favour the binding of ATP rather than the inhibitor. In NBL, for instance, the ALK F1174L point mutation alters the protein’s structure in a way that increases the ATP binding affinity.

On contrary, when type II kinase inhibitors bind the adenine pocket, they also induce a conformational change, which stabilizes the inactive state [78,79]. Furthermore, type II kinase inhibitors extend their binding to the adjacent hydrophobic pocket [58]. However, resistance towards kinase II inhibitors can be acquired by mutations in the kinase domain [78]. Finally, type I ½ inhibitors comprise features of both type I and type II kinase inhibitors: they bind the ATP binding pocket in its active state, similar to a type I kinase inhibitor, but they simultaneously bind the hydrophobic pocket in the same way as a type II kinase inhibitor [58,79]. The shape and size of the hydrophobic back pocket is mainly determined by the gatekeeper residue (L1196 in ALK); therefore, type I ½ and type II kinase inhibitors (Figure 3) can be adjusted to fit amino acids, both wildtype and mutated, at this position [79]. As stated before, the gatekeeper residue is rarely mutated in NBL.

In NBL, the ALK hotspot mutations are not part of the ATP binding site but are situated in the hydrophobic cavity. Therefore, ALK inhibitors with an extended binding area will likely be more efficient and less subjected to drug resistance.

The main advantage of type I ½ and II kinase inhibitors is their higher selectivity for ALK, i.e., side effects caused by off-target effects are reduced. IGFR1, for instance, has high sequence similarity to ALK and is therefore often an unwanted target. In contrast to ALK, IGFR1 is abundant in healthy tissue and is therefore likely to induce side effects [80]. There are further structural similarities among kinases and their binding pockets, contributing to the promiscuity of type I kinase inhibitors [79]. Furthermore, type I ½ and II inhibitors have lower disassociation rates and thus prolonged residence time, whilst they are also effective against most mutations [58,81].

Unfortunately, the higher selectivity comes with the drawback of a much lower binding affinity. These inhibitors are also larger and lipophilic molecules, as they both have to occupy a larger region and bind to the hydrophobic pocket. Thus, the cellular penetration of the inhibitors is impeded, reducing the efficiency of the drugs [81].

Regarding ALK, only one type I ½ inhibitor has been developed thus far: piperidine carboxamide [80], which has proven to be superior compared to both crizotinib and ceritinib, especially in respect to the point mutations ALK F1174L and R1275Q in in silico studies [58]. The binding of this type I ½ kinase inhibitor leads to dramatically conformational alterations, including the sites F1174 and R1275 in the hydrophobic cavity [7,58]. By analogue screening, compounds with even higher qualities have been identified, and a potent example is shown in Figure 3a. These molecules inhibit downstream signalling via Akt and STAT in addition to inhibiting ALK auto-phosphorylation and increasing anti-proliferative effects [58].

As type II inhibitors, pyrazolamine derivatives have been synthesized, and a prominent example is given in Figure 3b. These compounds have the additional potential to inhibit the putative auto-phosphorylation of tyrosine residues, which may affect downstream signalling. Unfortunately, these compounds have even lower kinase binding affinities than the type I ½ inhibitors [81].

In conclusion, type I ½ and II kinase inhibitors have high potential in treating amplified or mutated ALK and as a second-line treatment in cases that are resistant to the more classic competitive inhibitors. However, challenges regarding the binding affinity and solubility of the compounds will have to be addressed prior to clinical testing. In addition, type I ½ and type II inhibitors have mainly been synthesized as result from in silico (compound analogue screening) data thus far, where crystal structures of the bound and unbound ALK states are used as models. Therefore, the binding mechanisms of these compounds in vitro might not exactly comply with the models.

### 4.5. Other Inhibitors

Several other promising second-generation crizotinib/alectinib-derivates have been identified by drug screening tests. These inhibitors have the potential for improved efficiency on ALK targeting [7]. None of these are FDA approved; therefore, they will not be further discussed in this review.

Monotherapies against other receptors can indirectly inhibit ALK overexpression. Two examples of such drugs will be presented here: the inhibition of heat shock protein 90 (HSP90) and the use of differentiation agents such as retinoic acids.

#### 4.5.1. HSP90 Inhibition

Heat shock proteins are chaperones that also refold proteins after exposure to (thermic) stress. The geldanamycin (17-AAG) derivative alvespimycin (17-DMAG) has been studied in NBL cell lines [14]. The study investigated the efficacy of alvespimycin in the treatment of ultra-high-risk NBL, namely patients with both *MYCN* and ALK amplifications or *MYCN* amplifications in combination with the high-risk point mutation ALK F1174L. Alvespimycin resulted in cell cycle arrest and increased apoptosis in cell lines. Simultaneously, the expression of both ALK and N-Myc was reduced as was well as alvespimycin resensitized N-Myc overexpressing cells to crizotinib. Of interest, the effect of alvespimycin was the highest when both proteins were overexpressed [14]. The assumed mechanism behind this observation is that the cells become addictive to ALK and N-Myc overexpression, so the down-regulation of these proteins leads to decreased proliferation and disturbance of the cell cycle [14,82].

Of note, combination therapy of TAE-684 or ceritinib with HSP90 inhibitors also resensitized drug-resistant cells carrying the ALK F1174L point mutation [55].

#### 4.5.2. Cell Differentiation by Retinoic Acids

Retinoic acids, the most renowned being all-trans retinoic acid (ATRA), are vitamin A derivatives that are involved in cell differentiation and are already in use in the treatment of several malignancies, for instance as part of the consolidation phase of the NBL high-risk regimen. ATRA additionally inhibits cancer cell growth, and as an off-target-effect, downregulates *MYCN* [82]. In NBL, ATRA decreases ALK expression on the mRNA level, reducing the downstream phosphorylation of both Akt and MAPK. An increase in cancer cell apoptosis has been observed but only in cells with amplified ALK or ALK gain-of-function mutations such as the NBL hotspot mutations [82].

## 5. Combination Therapy in NBL Treatment

As illustrated in the presentation of ALK inhibitors in monotherapy (Section 4), single agents are rarely sufficient for obtaining remission/durable response in patients. This is mainly due to the acquisition of secondary mutations that lower the binding affinity of the inhibitors and due to pathway evasion strategies of the cancer cells.

Therefore, efforts have been made to identify suitable combination therapies. This chapter will focus on combinational therapies using ALK kinase inhibitors with downstream pathway inhibitors and ALK inhibitors in combination with other agents that target overexpressed ALK.

A caveat for the whole chapter (and also partly for Section 4.2) is that there are few published data on the various combination therapies; thus, the results have mostly not been validated in follow-up studies. Furthermore, all of the presented data are based on in vitro and in vivo experiments, and the results may therefore not precisely relate to the clinic. Another limitation is that there are few NBL cell lines available for particular ALK aberrations, and PDX experiments are usually restricted to one or two models due to significant time requirement and high costs, and xenografts tend to be obtained from only the most aggressive cell lines. Therefore, the results from both in vitro and in vivo experiments may not be representative of NBL in general.

In order to gain more evidence on the effect of combinatory therapy, results of individual studies should be validated in further experiments, and the studies extended to patient-derived cell cultures/PDX models. Clinical studies on combinatory therapies are therefore not yet impending.

### 5.1. Combinational Therapy with Pathway Inhibitors

#### 5.1.1. Inhibitors of the PI3K-Akt-mTOR Pathway

The PI3K-Akt-mTOR pathway is one of the three major ALK downstream signalling pathways (besides JAK-STAT and RAS/MAPK) and the most important pathway for gain-of-function point mutations [63] (Figure 2). Additionally, *MYCN* amplification and the overexpression of TrkB—two other factors associated with high-risk NBL—increase Akt-phosphorylation and thus signalling through this pathway [83,84].

Several promising results have been observed in pre-clinical studies. First, the PI3K-inhibitor pictilisib (GDC-0941) in combination with lorlatinib partly restored sensitivity towards the latter after acquired drug-resistance [74]. Second, the PI3K-inhibitor alpelisib in combination with ceritinib improved the effect of ceritinib in cell lines [85]. Most importantly, the dual PI3K-mTOR inhibitor gedatolisib (PKI-587) showed synergistic effects with crizotinib in ultra-high-risk NBL, i.e., *MYCN* amplifications in combination with the ALK F1174L mutation [86]. Crizotinib monotherapy down-regulated PI3K but not mTOR complex 1 (mTORC1) in cell lines harbouring *MYCN* amplifications [86]. The universal mTOR inhibitor Torin-2 effectively inhibited both mTOR complexes [87]; however, the loss of the feedback mechanisms rescued Akt phosphorylation in ultra-high-risk NBL [86]. Crizotinib in combination with the dual PI3K-mTOR inhibitor gedatolisib abrogated this Akt activation and also demonstrated effectiveness in less aggressive NBL, for instance, in ALK R1275Q mutated tumours [86]. Thus, targeting the PI3K-Akt-mTOR pathway at both ends appears to be a compelling approach.

#### 5.1.2. Inhibitors of the RAS/MAPK Pathway

The RAS/MAPK pathway is another important ALK downstream signalling pathway that leads to ALK inhibitor resistance in cell lines [74]. The mitogen-activated protein kinase (MEK1/2) inhibitor trametinib, while ineffective alone [47], blunted MAPK activation and increased lorlatinib sensitivity in a cell line study [74]. The supplementary combination with the PI3K inhibitor pictilisib showed even stronger effects [74]. Nevertheless, the inhibition of the RAS/MAPK pathway poses a risk of severe adverse effects that were demonstrated in another study: the MEK1/2 inhibitor trametinib in combination with the ALK inhibitor TAE-684 led to the upregulation of MAPK feedback inhibition. The downregulation of MAPK, however, is indicative of treatment resistance, disease progression, and adverse prognosis [47]. In NBL, adrenergic neural differentiation genes, which correlate with lorlatinib resistance (see Section 5.3), are also upregulated by a combinatory treatment of TAE-684 and trametinib [47]. Furthermore, MAPK inhibition is counteracted by upstream RAS/MAPK pathway mutations, which are more prominent at relapse [47,88]. Altogether, follow-up studies should address these contradicting results, which might indicate that inhibition of the RAS/MAPK pathway is not a practical strategy in NBL.

### 5.2. Combinatory Therapy with Other Agents

This chapter will focus on pre-clinical combinatory therapy using ALK inhibitors with agents against other targets, mainly other kinases, which directly or indirectly affect ALK.

#### 5.2.1. Inhibition of the Ubiquitin Kinase MDM2

The E3 ubiquitin kinase MDM2 deactivates the tumour suppressor TP53 by ubiquitination, and the upregulation of this kinase is linked with multidrug resistance [89].

Ceritinib combined with the selective MDM2 inhibitor CGM097 exerted synergistic anti-proliferative and pro-apoptotic effects [49]. In cell line studies, CGM097 resensitized cells to ceritinib, even when N-Myc was overexpressed. In PDX models, the combination led to complete and durable tumour regression; however, this only lasted as long as treatment persisted [49].

A premise for the treatment with GCM097 is that *p53* must not harbour point mutations. Almost all de novo NBL cases are wt *p53*, while at relapse, 85% of patients still do not carry *p53* mutations [49].

#### 5.2.2. Inhibition of Histone Deacytylases

The overexpression of histone deacytylases (HDAC) appears to relate to poor NBL prognosis. In that respect, the HDAC inhibitor valproate decreases proliferation while increasing apoptosis and cell differentiation in NBL cells [90]. Alectinib and the HDAC inhibitor vorinostat synergistically inhibit cell growth for ALK R1275Q mutated cells [91]. In this combination, apoptosis was induced, while N-Myc expression and signalling via the nuclear factor kappa B (NFκB) pathway were repressed [91]. The presented results were only based on one cell line.

#### 5.2.3. Inhibition of Cyclin Dependant Kinases

As pointed out earlier (Section 4.2.2), ALK F1174 point mutations are resistant towards crizotinib treatment. The combination of the latter with the dual cyclin dependant kinases 4 and 6 (CDK4/6) inhibitor ribociclib had an additive or a synergistic effect, respecitively, on a total of nine cell lines carrying ALK mutations, including the three NBL hotspot point mutations [85]. Interestingly, there was no response to treatment with ribociclib alone. The combinatory treatment decreased ALK-phosphorylation and induced cell-cycle arrest via phosphorylation of the retinoblastoma protein (Rb), which appears to be downstream ALK [85]. The same trend was observed in mural models, where the ALK R1275Q mutation proved to be the most sensitive; ceritinib may account for this difference since ALK R1275Q is the most ceritinib-sensitive mutation [85]. Even though the results have not been validated in follow-up studies, the authors included a total of 17 cell lines— several of each hotspot point mutation—which adds to the significance of the potential of combining ALK and CDK4/6 inhibitors.

#### 5.2.4. Inhibition of the Axl Kinase

Axl is a receptor tyrosine kinase that plays a prominent role in the nervous system [92] and regulates epithelial-to-mesenchymal transition (EMT) [93]. It induces proliferation while repressing apoptosis via the RAS/MAPK and PI3K-Akt-mTOR pathways [55,93]. Axl is already acknowledged as a mediator of resistance towards tyrosine kinase inhibitors [80,94]. In respect to this, ligand-dependant Axl activation seems to be involved in poor drug response towards the ALK inhibitors crizotinib and ceritinib [93]. Specifically, the ALK F1174L point mutation drives TAE-684 and ceritinib resistance in NBL cell lines [55]. The study on NBL further showed that Axl upregulation bypasses ALK inhibitor-induced MAPK downregulation and induces EMT [55]. TAE-684/ceritinib resistant cell lines harbouring the ALK F1174L point mutation could be resensitized to the drugs upon the inhibition of the Axl ligand HSP90 (see Section 4.5.1) [55]. A weakness of this study is that the experiments were only conducted on one cell line and its derivative (both ALK F1174L mutated). On the contrary, a follow-up study showed that even though caspase-3 induced apoptosis was observed in the cell lines, no anti-tumoral effects in PDX models were evident when crizotinib or ceritinib were combined with the Axl inhibitor bemcentinib (BGB324) [93]. However, the results of the latter study were also based on only one cell line.

#### 5.2.5. Inhibition of the Wnt/β-Catenin Pathway

The Wnt/β-catenin pathway has shown to be involved in treatment resistance towards ALK inhibitors [50]. A recent cell line study showed that both the alectinib and Wnt inhibitors decreased cell viability, and in combination, they displayed an additive effect [95]. Additionally, a PDX study indicated Wnt/β-catenin pathway activation in aggressive prostate cancer, e.g., a cancer harbouring both an *MYCN* amplification and an ALK F1174C point mutation. The combination of alectinib and the Wnt inhibitor ICG-001 was able to suppress tumour metastases [95]. Thus, these data indicate that the Wnt/β -catenin pathway is involved in tumour growth and survival, which might also be valid for NBL.

### 5.3. On the Verge—The Combination of ALK Inhibitors with TRAIL

Recent studies on cell lines and PDX models introduce an innovative new approach in dealing with ALK inhibitor resistance.

As mentioned in the NBL background section (Section 2.1), NBL tumours are heterogeneous and consist of both undifferentiated MES cells and lineage-committed ADRN cells. During chemotherapy, MES cells are selected for, because they are less chemosensitive [96] and also because EMT is an evasion strategy upon ALK inhibition [55]. and because EMT is an evasion strategy upon ALK inhibition [55]. In concordance with that, preclinical studies of Westerhout and van Nes [97,98] indicate that MES cells drive relapse, as only ADRN cells express ALK on the mRNA and protein levels and thus are sensitive towards ALK inhibitors. The transition of ADRN cells to MES cells by EMT further decreases drug-response.

The group therefore combined lorlatinib with tumour necrosis factor-related apoptosis inducing ligand (TRAIL), which led to caspase-8 induced apoptosis in MES cells without affecting the ADRN cells. In PDX models, the combinatory treatment of lorlatinib with TRAIL delayed tumour relapse [97,98]. This drug combination strategy directly targets the biology of NBL tumours, which is an inspirational approach that needs to be evaluated further.

## 6. Conclusions

Therapeutic targeting of the ALK kinase is a promising strategy for the approximately 14% of de novo NBL patients that carry ALK aberrations. There are few outcomes from clinical trials that are available, as several studies are still recruiting or have not been initiated yet. However, data from completed clinical trials of first- and second-generation ALK inhibitors demonstrate the limitations of these drugs for the achievement of a durable response. Furthermore, pre-clinical studies of kinase type I inhibitors in NBL in general point towards the risk of acquired drug resistance. Lasting effects are mainly observed for patients carrying the ALK R1275Q point mutation, which has the lowest ATP-binding affinity of the three NBL hotspot mutations. For these patients, ALK inhibitors may become part of the standard consolidation treatment. Furthermore, newer generations of ALK inhibitors have increased effects on the more aggressive point mutations in vitro, but clinical trials have yet to demonstrate their potency in a real-life setting.

It is not unlikely that it will be necessary to combine ALK inhibitors with other inhibitors/drugs that either target downstream pathways—targeting the PI3K-Akt-mTOR pathway appears to be an especially promising strategy—or that otherwise influence ALK degradation, ALK phosphorylation, and cancer cell proliferation and apoptosis in general.

ALK-specific type kinase I ½ and II inhibitors may arise as especially effective drugs, as they keep ALK in its inactivated state without the risk of severe off-target effects. Type II kinase inhibitors, such as imatinib, have already shown to be very potent in the treatment of other malignancies [99]. Furthermore, drug combinations that target the heterogeneous biology of NBL lesions—such as ALK inhibitors together with TRAIL—are intriguing strategies that should be investigated more thoroughly.

Altogether, more research is needed to identify the most effective drugs/drug combinations for the different types of ALK aberrations. Such drugs, in combination with standard chemotherapy or as consolidation treatment, might employ the potential to induce durable remission or stable disease for the subgroup of high-risk NBL patients harbouring ALK mutations.

## Figures and Tables

**Figure 1 pharmaceutics-13-01427-f001:**
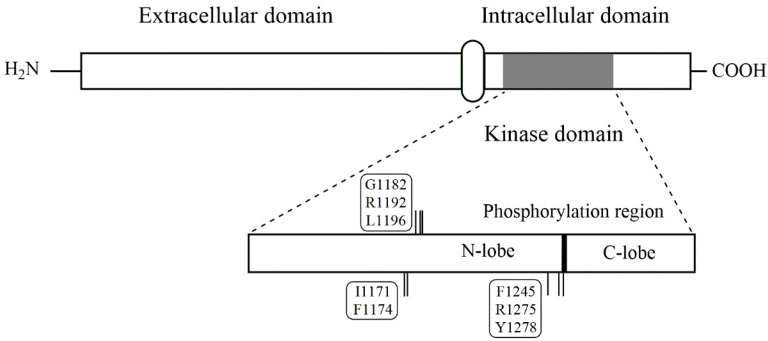
Schematic overview of the ALK protein. In the upper part of the figure, the extracellular domain, the transmembrane segment and the intracellular domain are shown. In the lower part, the kinase domain is enlarged showing the positions of the point mutations described in the text.

**Figure 2 pharmaceutics-13-01427-f002:**
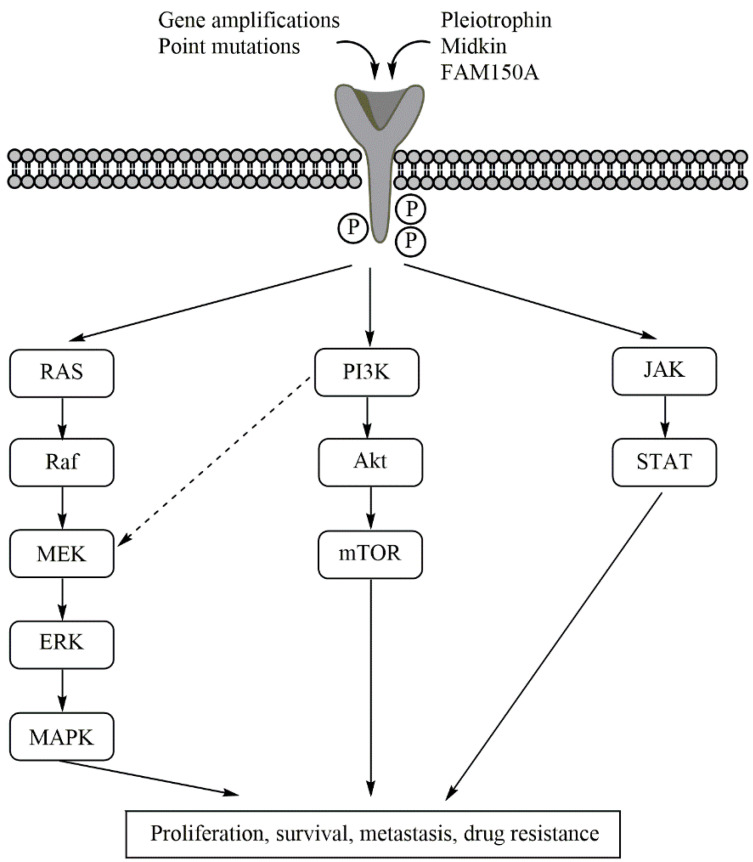
The downstream ALK activation pathways. ALK is either activated by its ligands pleitrophin, midkin, and FAM150A, or by ALK aberrations. The figure highlights the pathways activated in aberrant ALK, where these pathways also contribute to drug evasion effects. Upon ALK auto-phosphorylation, the RAS/MAPK, PI3K/Akt/mTOR, and JAK/STAT pathways are activated. Signalling through these pathways is correlated with increased cancer cell proliferation, decreased cell apoptosis, metastization, and drug resistance. The interconnection between the RAS/MAPK and PI3K/Akt/mTOR pathways is also indicated.

**Figure 3 pharmaceutics-13-01427-f003:**
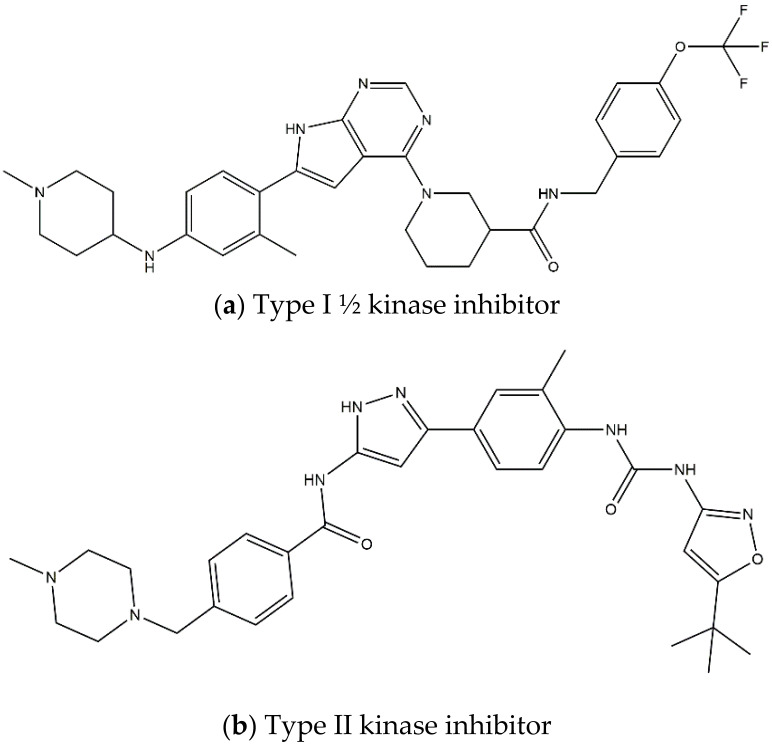
Structures of prominent candidates for a (**a**) type I ½ and a (**b**) type II anaplastic lymphoma kinase inhibitor as presented in [7].

**Table 1 pharmaceutics-13-01427-t001:** Important ALK point mutations that are described in the text. The gatekeeper mutation and the mutations associated with drug resistance, both towards specific inhibitors and conventional chemotherapy, are uncommon in neuroblastoma.

Specific Positions	Residues
Auto-phosphorylation site	Y1278, Y1282, Y1283
NBL germline mutations	G1182, R1192, R1275
NBL hotspot mutations	F1174, F1245, R1275
Less common NBL mutations	I1170, I1171, Y1278
ALK gatekeeper mutation	L1196
Mutations involved in drug resistance	G1202, G1269

**Table 2 pharmaceutics-13-01427-t002:** Characteristics and structures of first-, second-, and third-generation of anaplastic lymphoma kinase inhibitors.

**First-Generation Inhibitor**			
**Crizotinib** Off-targets: ROS1, c-MET No CNS penetration Sensitive: R1275Q Resistant: F1174L/V	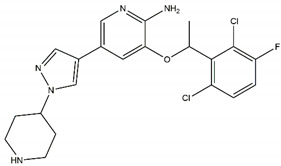		
**Second-Generation Inhibitors**			
**TAE-684** Sensitive: R1275Q, F1174L Discontinued	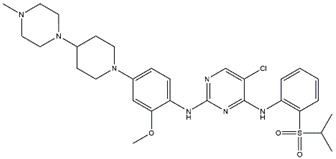	**Ceritinib** Off-targets: IGFR1, ROS1 CNS penetration Resistant: F1174L/C	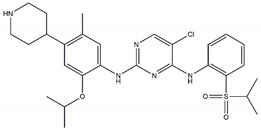
**ASP3026** Off-targets: ROS1, ACK Resistant: L1196M	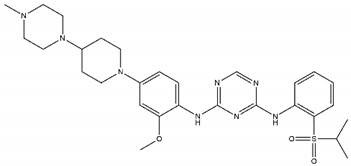	**Brigatinib** Off-targets: ROS1, EGFR CNS penetration Sensitive: all NBL-associated point mutations	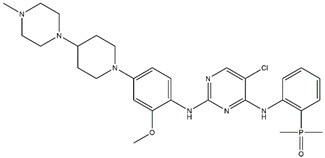
**Ensartinib** Sensitive: R1275Q, F1174L	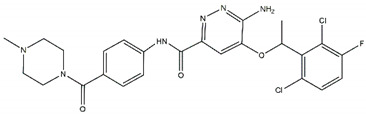	**AZD3463** Off-target: IGFR1 Sensitive: NBL hotspot mutations	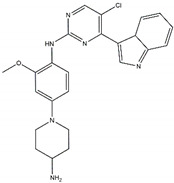
**CEP-37440** Off-target: FAK CNS penetration Sensitive: NBL hotspot mutations	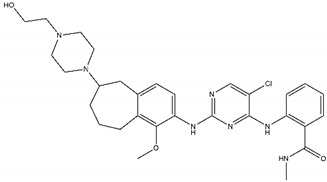	**Alectinib** Sensitive: all NBL-associated point mutations, plus amplifications	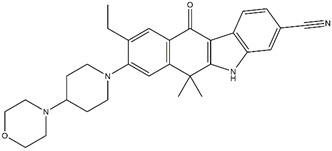
**Belizatinib** Off-targets: IGFR1, JAK2, TrkA/B/C, c-Src CNS penetration Sensitive: R1275Q, L1196M	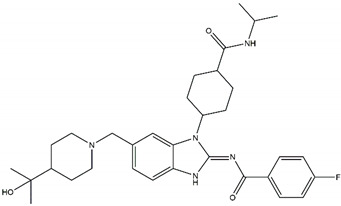	**Entrectinib** Off-targets: ROS1, TrkA/B/C CNS penetration Sensitive: ALK amplifications	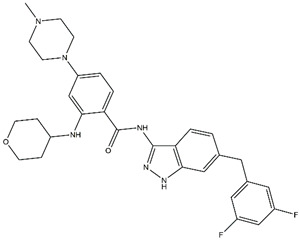
**Third-Generation Inhibitors**			
**Lorlatinib** Off-target: ROS1 CNS penetration Sensitive: NBL hotspot mutations	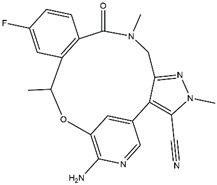	**Repotrectinib** Off-targets: ROS1, TrkA/B/C CNS penetration Sensitive: all NBL-associated point mutations	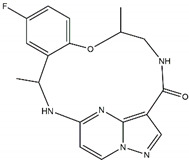

Abbreviations: ROS1: ROS pro-oncogene 1; c-MET: hepatocyte growth factor; CNS: central nervous system; IGFR1: insulin-like growth factor; ACK: activated CDC24 kinase; EGFR: epidermal growth factor receptor; NBL: neuroblastoma; FAK: focal adhesive kinase; JAK2: Janus kinase 2; Trk: tropomyosin receptor kinase.

**Table 3 pharmaceutics-13-01427-t003:** ALK inhibitors in clinical trials that include NBL patients.

Drug	Phase	Inclusion Criteria	Study Identifier	State	Published Data
Crizotinib	I/II	Refractory NBL	NCT00949770	Completed	[32,44]
	I	Malignant neoplasms	NCT01121588	Active	
	III	High-risk NBL	NCT03126916	Recruiting	
Ceritinib	I	ALK-activated tumours	NTC01742286	Completed	[53]
	I	Relapsed/refractory NBL	NTC02780128	Recruiting	
	II	*De novo* NBL	NTC02559778	Recruiting	
ASP3026	I	Crizotinib-resistant tumours	NCT01284192	Completed	[54]
Ensartinib	II	Relapsed NBL	NTC03213652	Recruiting	
	II	Relapsed/refractory NBL	NTC03155620	Recruiting	
Entrectinib	I/II	ALK-activated tumours	NTC02650401	Active	
Lorlatinib	I	Relapsed/refractory NBL	NCT03107988	Recruiting	
	I	*De novo* NBL	NCT04753658	Recruiting	

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
