# Peer review of "Therapeutic Targeting of the Anaplastic Lymphoma Kinase (ALK) in Neuroblastoma—A Comprehensive Update"

_pharmaceutics, 2021, doi:10.3390/pharmaceutics13091427_

Round 1
Reviewer 1 Report
The manuscript pharmaceutics-1327257, Therapeutic targeting of the anaplastic lymphoma kinase (ALK) in neuroblastoma – a comprehensive update, presents a very interesting and valuable collection of data on ALK inhibitors and their therapeutical potencial. The paper is very well-structured and seems to be an exhaustive presentation on the proposed theme.
The manuscript lacks a proper introduction in which the authors should present the objective of this review and the type of data that are analyzed. The authors should argue why is this review important, if there are any similar work they should be referenced and shortly discussed highlighting the originality of this new work.
The manuscript lacks the line numbers making the review process difficult. On page 3, in section 2.1., the authors should better detail on the structure of the PK, as they later discuss “gatekeeper mutation” and ATP-binding pocket.
In the section 2.1 the authors should prepare a scheme with ALK signaling pathway.
The section 2.2 could use a table to present major mutation of ALK that would make easier the reading of the section 3.1.
The authors should carefully check all the structures in table 1. Because of the editing, some of them are trimmed down and some sectioned are not visible. ALL the structures should be checked for mistakes. For example, crizotinib is presented as an imidazole derivative whereas it should be pyrazole. Lorlatinib is missing the fluor atom, which is very important considering it evolved as a cyclic derivative of crizotinib.
The authors are introducing the concepts kinase type I inhibitors or types II, but without detailing the authors what this means. There is a series of reviews by Robert Roskoski on approved small molecule protein kinase inhibitors that I recommend for the authors. There is an article each year and all of them could be a very good model to improve this review.
In the section where the authors discuss combinations of therapies, it would be helpful to add doses for each compound mentioned.
I would also recommend the authors to have a more critical view on their data. The review should not be just a collection of data. The authors should present also contradicting data and new hypotheses. Did the authors detected some mistakes in any of the papers reviewed or some major shortcomings? This is the difficult part when writing a review paper, but it is what add true value to it.
Considering the high volume of work done in this field, the number of references could be higher.
Author Response
The manuscript pharmaceutics-1327257, Therapeutic targeting of the anaplastic lymphoma kinase (ALK) in neuroblastoma – a comprehensive update, presents a very interesting and valuable collection of data on ALK inhibitors and their therapeutical potencial. The paper is very well-structured and seems to be an exhaustive presentation on the proposed theme.
Thank you for your kind response. We especially would like to thank you for your constructive advises. In our opinion they helped us in improving the manuscript. And thank you for being thorough, like spotting errors in the chemical structures.
All the changes in the text are highlighted in the revised version of the manuscript.
The manuscript lacks a proper introduction in which the authors should present the objective of this review and the type of data that are analyzed. The authors should argue why is this review important, if there are any similar work they should be referenced and shortly discussed highlighting the originality of this new work.
We agree that the former introduction section was abrupt. We added a short introduction (lines 25-39) about the objective of the study, important work the review is based upon, and a statement of the novelty of the article, i.e. a summary of proposed combinatory therapies.
The manuscript lacks the line numbers making the review process difficult. On page 3, in section 2.1., the authors should better detail on the structure of the PK, as they later discuss “gatekeeper mutation” and ATP-binding pocket.
The section 2.2 could use a table to present major mutation of ALK that would make easier the reading of the section 3.1.
We included a scheme (Figure 1) of ALK, including the positions of discussed amino acids. The table below also explains the nature of the presented amino acids. We hope this choice of resolving the two advises above are acceptable.
In the section 2.1 the authors should prepare a scheme with ALK signaling pathway.
As also requested by reviewer 3, we added a Figure 2 to illustrate the downstream pathways of activated ALK, which are of special importance in NBL.
The authors should carefully check all the structures in table 1. Because of the editing, some of them are trimmed down and some sectioned are not visible. ALL the structures should be checked for mistakes. For example, crizotinib is presented as an imidazole derivative whereas it should be pyrazole. Lorlatinib is missing the fluor atom, which is very important considering it evolved as a cyclic derivative of crizotinib.
Thank you for checking the structures for errors. A total of five molecules have been edited.
The authors are introducing the concepts kinase type I inhibitors or types II, but without detailing the authors what this means. There is a series of reviews by Robert Roskoski on approved small molecule protein kinase inhibitors that I recommend for the authors. There is an article each year and all of them could be a very good model to improve this review.
In our opinion, an entire section about different kinds of kinase inhibitors would be beyond the scope of this review. However, based on the reviews by Robert Roskoski, we included the original articles on the nature of type I ½ (Zuccotta et al.) and type II (Dar & Shokat) kinase inhibitors, and extended the explanation of these inhibitors (lines 395-405 and lines 412-414). We hope these changes are acceptable.
In the section where the authors discuss combinations of therapies, it would be helpful to add doses for each compound mentioned.
There are several reasons, why we decided not to include doses:
All experiments on combinatory therapies were conducted in cell line and/or PDX models. Different cell lines usually require different dosages due to differing sensitivities. In our opinion, there should be more studies that also include more cell lines (see comment below) to conclude whether the required dosages are sensible.
Furthermore, results from cell line and PDX models cannot directly be transferred to the clinic, where combination therapies likely also will have to include standard chemotherapy. Toxicities probable will depend on cross-interaction between different drugs.
But, of course, these issues will have to be further addressed before dose escalation studies on patients can be conducted.
I would also recommend the authors to have a more critical view on their data. The review should not be just a collection of data. The authors should present also contradicting data and new hypotheses. Did the authors detected some mistakes in any of the papers reviewed or some major shortcomings? This is the difficult part when writing a review paper, but it is what add true value to it.
A reasonable comment. Researchers tend to not wanting to discredit other researchers’ work. We indeed identified flaws in several studies, especially in the studies on combinatory therapies.
There are general shortcomings in that there are very few studies available on different drugs/combinations and that, in general, few cell lines were used. We therefore added a section about these shortcomings in the start of chapter 5 (lines 486-498). Furthermore, we added comments to several studies where only one model was used (highlighted throughout section 5.1 and 5.2). We also stressed the conflicting results on inhibiting the RAS/MAPK pathway, which might be a severe limitation of this approach (lines 526-527 and 533-535).
On the other hand, we also compliment a study that included a large number of cell lines, and where the results therefore are more indicative of a real observed effect (lines 568-571).
We hope this alterations are acceptable for the reviewer.
Considering the high volume of work done in this field, the number of references could be higher.
We agree that 100 citations is a relatively small number for a review. There are mainly two reasons for that:
There are few studies on the different inhibitors (alone and in combination) published. Therefore, sections 4 and 5 contain relatively few citations. That is a weakness for the evidence of the published results, which now is also stated as discussed above.
Section 2 and 3, on the other hand, give a general background on neuroblastoma and the ALK kinase. These are mostly basic knowledge within the field, where it often proves to be difficult to identify the original source. Recent updates in the field were included like current risk stratification, treatment regimes and the variety of ALK inhibitors.
Reviewer 2 Report
The authors described the importance of targeting ALK in neuroblastoma. This article is well-constructed and is very informative for oncologists. Therefore, it is acceptable to out journal. However, the reviewer requests the authors to describe more precisely the relationship between MYCN and ALK or the regulation of each other.
Author Response
The authors described the importance of targeting ALK in neuroblastoma. This article is well-constructed and is very informative for oncologists. Therefore, it is acceptable to out journal. However, the reviewer requests the authors to describe more precisely the relationship between MYCN and ALK or the regulation of each other.
Thank you for your kind reply. We extended the section on the relationship between MYCN and ALK (lines 142-151). We hope this change is acceptable.
All other changes in the manuscript, as required by the other reviewers, are highlighted in the text.
Reviewer 3 Report
This manuscript comprehensively reviews the structure, function, mutation, and therapeutic role of ALK in neurobalstoma. It describes the potential and limitation of ALK inhibitors in clinical use in high-risk neuroblastoma and focuses on the combination of ALK inhibitors with drugs that target downstream signaling pathways to overcome the development of acquired resistance to ALKi.
Overall, it is a well-written review. The minor suggestion from the reviewer is that the addition of cartoons to illustrate the structure and the oncogenic role of ALK in NBL and the compensatory pathways for the development of acquired resistance to ALK inhibitors will be greatly helpful for the readers
Author Response
Overall, it is a well-written review. The minor suggestion from the reviewer is that the addition of cartoons to illustrate the structure and the oncogenic role of ALK in NBL and the compensatory pathways for the development of acquired resistance to ALK inhibitors will be greatly helpful for the readers.
Thank you for your kind reply. We included a schematic illustration (Figure 1) that shows the basic structure of ALK, where the residues that are mentioned in the text are shown and explained. Reviewer 1 also asked for structure and overview of ALK, so we hope that this compromise is acceptable for both reviewers.
We furthermore added a Figure 2 to illustrate the downstream pathways of activated ALK, which are of special importance in NBL.
All other changes in the manuscript, as required by the other reviewers, are highlighted in the text.
Round 2
Reviewer 1 Report
The authors responded to all the comments and suggestions and clearly argued if some modifications were not performed. The manuscript has considerably improved and I congratulate the authors for it.